# A Pilot Analysis on the Efficacy of Multiple Trigger-Point Saline Injections in Chronic Tension-Type Headache: A Retrospective Observational Study

**DOI:** 10.3390/jcm11185428

**Published:** 2022-09-15

**Authors:** Sung-Cheol Cho, Dong-Rak Kwon, Jeong-Won Seong, Yuntae Kim, Levent Özçakar

**Affiliations:** 1Department of Rehabilitation Medicine, School of Medicine, Catholic University of Daegu, Daegu 42742, Korea; 2Department of Family Medicine, Sarang Tong-sa Research Center, Jinju City 52686, Korea; 3Department of Physical Medicine and Rehabilitation, Soonchunhyang University Cheonan Hospital, College of Medicine, Soonchunhyang University, Cheonan 31151, Korea; 4Department of Physical and Rehabilitation Medicine, Medical School, Hacettepe University, 06800 Ankara, Turkey

**Keywords:** head, neck, pain, muscle, intervention, ultrasound

## Abstract

This study aimed to investigate the efficacy of new targeted trigger-point injections (TPIs) using isotonic saline in patients with chronic tension-type headache (CTTH). Of 121 patients with headache who were retrospectively reviewed, 19 were included in this study and were categorized into two groups: those who received TPIs more than four times (group 1); and those who received TPIs less than, or equal to, four times (group 2). The patients received ultrasound-guided isotonic saline injections into the active trigger points once weekly. The primary outcome was an effect on headache intensity, determined using the visual analog scale (VAS), whereas the secondary outcome was an effect on quality of life, evaluated using the Henry Ford Hospital Headache Disability Inventory (HDI). The mean symptom duration of the 19 patients (11 men and 8 women; mean age, 52.5 years; and range, 23–81 years) was 16 months. The most frequently injected muscle was the splenius capitis. Patient demographics were similar between the two groups (*p* > 0.05). Simple linear regression revealed that symptom duration (*p* = 0.001) and baseline VAS score (*p* = 0.009) were significantly associated with the number of injections. At one month after the first injection, the mean VAS and HDI scores in group 2 were significantly lower than those in group 1 (*p* < 0.05), whereas the scores significantly decreased immediately after the last injection in both groups (*p* < 0.05). No adverse effects were reported in any patient. Our results indicate that the administration of new targeted TPIs using isotonic saline into the head and neck muscles of patients with CTTH can effectively relieve headache intensity and safely improve their quality of life.

## 1. Introduction

Among the most common symptoms of neurological disorders, headache, characterized by pain in the head and face, is experienced by >90% of people worldwide. Chronic headache is diagnosed when frequent episodes occur that last for ≥15 days per month or 180 days per year, persisting for several months or years. It affects 4–5% of the population and is considered one of the most common reasons to visit the doctor [1,2,3]. In general, headaches are classified according to the criteria reported in the third edition of the International Classification of Headache Disorders (ICHD-3) [4].

Tension-type headache (TTH), previously known as muscle-contraction headache, is characterized by the presence of mild-to-moderate pain that occurs continuously or episodically. TTH often occurs bilaterally, with a “pressing” or “tightening” sensation. It is not associated with nausea or vomiting, but may be associated with photophobia or phonophobia on rare occasions [5]. TTH is the most common type of headache disorder in adults. The estimated prevalence of TTH and chronic TTH (CTTH) has been reported to be 38.3% and 2.2%, respectively [6].

To date, the pathophysiology underlying chronic headaches remains poorly understood. The peripheral muscles in the head and neck are critically important for their development [7,8]. Other important factors include the sensitization of pain transmission circuits at the trigeminal nucleus, nociception from the pericranial muscles, and dysregulation of central pain modulation [5,9]. Additionally, the cervical muscles induce an important mechanism associated with chronic pain—central sensitization—through the following regions: medullary dorsal horn and upper cervical (C1 and C2) dorsal horn [10]. For instance, the muscles around the head and neck, such as the sternocleidomastoid, upper trapezius, splenius capitis (SPC), and temporalis muscles, have been reported to cause CTTH [11].

Currently, a standard treatment protocol for CTTH is not available due to poor understanding of its mechanism. So far, numerous pharmacological and nonpharmacological therapies have been performed. Notably, trigger-point injections (TPIs) with local anesthetics or corticosteroids are commonly used for the treatment of TTH; however, repeat injections tend to induce toxicity to the muscles [12,13,14]. A recent study on patients with myofascial pain syndrome has demonstrated that the therapeutic effects (on pain reduction and functional recovery) were similar between a group injected with isotonic saline and another group injected with a mixture of lidocaine and corticosteroids [15].

Therefore, the purpose of this study was to investigate the efficacy of new targeted TPIs using isotonic saline, which does not elicit side-effects with repeat injections. We also aimed to clarify the relationship among clinical features, CTTH symptoms, and the frequency of TPIs.

## 2. Materials and Methods

### 2.1. Design and Setting

The study approval was granted by the Institutional Review Board (IRB) and the Ethics Committee at the Daegu Catholic University Hospital (IRB no.: CR-20-047). The study protocol was executed in accordance with the Declaration of Helsinki. The need for informed consent was waived, owing to the retrospective nature of the study.

The medical records of outpatients who had visited the Department of Physical Medicine and Rehabilitation at the Daegu Catholic University Medical Center for chronic headache between 2018 and 2020 (*n* = 121) were reviewed (Figure 1).

### 2.2. Patient Characteristics

In total, 121 patients had not responded to previous conservative therapies (e.g., medication, physical therapy) that were used to treat headaches. The inclusion criteria were as follows: (1) aged 19–65 years, (2) positive diagnosis of episodic CTTH as per the ICHD-3 beta version [4], (3) presence of head and neck myofascial trigger points that can be identified to reproduce the patient’s headache pain, and (4) absence of other significant pain problems. Patients presenting with secondary headaches (e.g., trauma, tumor) or those having other conditions that may be accompanied by headaches, including fibromyalgia, diabetes, depression, nervous system or cardiovascular disease, and pregnancy, were excluded from the current study. Patients were subsequently categorized into two groups according to the injection frequency: those who had received TPIs more than four times (group 1) and those who had received TPIs equal to, or less than, four times (group 2).

### 2.3. Intervention

Patients were asked to visit the outpatient clinic every week after their first visit. The medical history of each patient was recorded and a physical examination was performed by a physiatrist during every visit. The affected muscles and associated trigger points were identified according to the following diagnostic criteria: (1) the presence of a palpable taut band of the skeletal muscle, (2) the presence of a hypersensitive tender spot in the taut band, (3) a local twitch response stimulated by the snapping palpation applied across the taut band, and (4) a recurrence of the typical type of referred pain at the trigger point resulting from compression [16].

Under ultrasound (US) guidance, isotonic saline injections were injected into the active trigger points identified during the physical examination (Figure 2). US-guided injections were administered by a physiatrist specialized in musculoskeletal US (for 17 years) using a 9–4 MHz multifrequency linear transducer (EPIQ 5, Philips Healthcare, Bothell, WA, USA). Passive muscle stretching was performed at the injection site after each TPI.

### 2.4. Data Collection and Outcome Measures

The characteristics and intensity of the headaches were assessed at each visit using a 10 cm horizontal visual analog scale (VAS), ranging from 0 (“no pain”) to 10 (“the worst imaginable pain”). In this study, the mean amount of pain experienced by patients over a period of 24 h before the assessment was considered to determine their VAS scores [17].

Furthermore, the patients completed the Henry Ford Hospital Headache Disability Inventory (HDI) questionnaire during each visit. The HDI consists of 25 items to assess the functional disabling effects of headache. Each item can be answered as “yes” (4 points), “sometimes” (2 points), or “no” (0 points). The highest score of the questionnaire can be 100 points, and a higher score would imply being more severely affected by headache in daily life [18]. The intensity of headache (VAS scores) was set as the primary measured outcome, whereas the quality of life (HDI scores) was set as the secondary measured outcome. Additionally, the demographic data of the patients were collected.

### 2.5. Statistical Analysis

IBM SPSS ver. 19.0 (IBM Co., Armonk, NY, USA) was used for statistical analyses, and significance was indicated by *p* < 0.05. The chi-squared test, Mann–Whitney U test, and Wilcoxon signed-rank test were used to determine the inter- and intra-group statistical differences. The correlations among age, number of injections, symptom duration, and baseline VAS score were analyzed using linear regression.

## 3. Results

In total, 121 charts were reviewed to select the 19 patients included in this study (11 males and 8 females; mean age, 52.5 years; and range, 23–81 years), who showed a mean symptom duration of 16 months (Table 1). The median number of injections was 9 (range, 3–29).

The demographic data of the two groups were similar (*p* > 0.05, Table 1). The SPC muscle was the most frequently injected site (Table 1).

Compared with the baseline score of 6.31 ± 2.13, the VAS score decreased to 2.57 ± 1.53 after the last injection (*p* < 0.05) in 19 patients. At 1 month after the first injection, the mean VAS score in group 2 (2.86 ± 1.57) was significantly lower than that in group 1 (5.25 ± 1.96; *p* < 0.05, Table 2). Moreover, the differences between VAS scores at baseline and 1 month after the first injection in group 2 were significantly lower than those in group 1 (*p* < 0.05, Table 2). However, the two groups had similar VAS scores immediately after the final injection as well as similar differences between the scores at baseline and immediately after the final injection (*p* > 0.05, Table 2).

Compared with the baseline score of 59.05 ± 26.80, the HDI score decreased to 28.15 ± 18.24 after the final injection (*p* < 0.05) in 19 patients. At 1 month after the first injection, the mean HDI score in group 2 was significantly lower than that in group 1 (*p* < 0.05, Table 2). Moreover, the differences between HDI scores at baseline and 1 month after the first injection in group 2 were significantly lower than those in group 1 (*p* < 0.05, Table 2). However, the two groups had similar HDI scores immediately after the final injection as well as similar differences between scores at baseline and immediately after the final injection (*p* > 0.05, Table 2).

Simple linear regression analyses revealed that symptom duration (*p* = 0.001) and baseline VAS score (*p* = 0.009) were significantly associated with the number of injections (Figure 3). Adverse effects or events were not reported in any patient.

## 4. Discussion

The present study aimed to explore the efficacy of multiple TPIs by injecting isotonic saline into the head and neck muscles of patients with CTTH. Our results demonstrated that the injected patients experienced decreased headache intensity and improved quality of life. Furthermore, the symptom duration and baseline VAS scores were significantly associated with the number of TPI injections. To the best of our knowledge, this is the first study to investigate the effectiveness of TPIs using saline for treating CTTH.

In this study, the VAS and HDI scores in group 2 were found to significantly decrease one month after the first injection compared with those in group 1. However, no intergroup differences were observed immediately after the final injection. In addition, this study showed that the therapeutic effect was highly associated with the symptom duration and baseline VAS scores. Hence, the clinician can identify the symptom duration and pain severity to predict the treatment outcome, which reassures patients and increases treatment compliance.

TTH is the most prevalent type of primary headache, and CTTH has been reported to affect 4–5% of the population [19]. Numerous pharmacological and nonpharmacological treatments, including the administration of nonsteroidal anti-inflammatory drugs, such as analgesics, antidepressants, and antispasmodics, are ineffective in treating CTTH. These drugs are typically used as abortive pharmacological medications [20].

Similar to other painful musculoskeletal disorders, CTTH can be treated using TPIs through myofascial trigger points, i.e., hyperirritable spots in the muscle. Our study demonstrated that the SPC muscle was the most frequently injected site. The injection sites indicated in our study are unlike those in previous studies, which have reported that head or neck muscles, such as the upper trapezius, sternocleidomastoid, and temporalis, are associated with CTTH [21]. This is, possibly, the first study that has examined the potential of SPC TPI injections in patients with CTTH. The SPC is a pericranial muscle that originates from the spinous processes of C7–T4 vertebrae. The insertion points are directed upward and laterally, extending to the occipital bone immediately below the lateral one-third of the superior nuchal line as well as the mastoid process of the temporal bone and below the sternocleidomastoid muscle; thus, SPC displays the strongest physical torsion, causing increased number of muscle tender points [22].

Several reports have been published on the efficacy of TPI as a treatment option for headache. In most of these studies, corticosteroid or lidocaine were used as the injectates. Baron et al. [13] injected a mixture of 1 mL (6 mg) of betamethasone sodium phosphate and 2 mL of 0.25% bupivacaine hydrochloride into the muscles of 147 patients with cervicogenic headache and reported symptom improvement in 88% of the patients immediately after the injection. Another study investigating the effects of lidocaine injections into the head and neck muscles (i.e., temporalis, SPC, and sternocleidomastoid) of patients with CTTH reported a significant improvement in the intensity and frequency of headache after three sessions applied every three days [12].

However, repeated injections of local anesthetics or corticosteroids may cause toxicity to the muscles in addition to several other side-effects. For example, lidocaine, one of the most commonly used local anesthetics, may cause cardiovascular problems, such as high or low blood pressure, and arrhythmia. Further, lidocaine injections can lead to neurological problems, including dizziness, blurred vision, and visual disturbance [23]. Likewise, frequent local injections of corticosteroids can lead to endocrinological and cardiovascular adverse effects, as well as muscle weakness [24].

In this study, multiple TPIs with isotonic saline were administered to patients with CTTH. The median number of injections was nine, and no adverse effects were observed. The primary measured outcome was headache intensity, and the VAS scores significantly improved after the injections. Furthermore, the secondary measured outcome—quality of life—significantly improved after the injections. Although this study lacked a control group, our results may encourage the administration of multiple TPIs (with isotonic saline) as a safe and cost-effective option for treating CTTH. Supporting our results, a study on myofascial pain syndrome showed that pain reduction and functional recovery following the administration of TPIs were similar in different groups treated with lidocaine, corticosteroids, or isotonic saline [15]. Tschopp et al. [25] reported similar results among patients presenting with facial pain and trigger points in the masticatory muscles, who were treated with 1 mL of 0.25% bupivacaine, 1% lidocaine, or 0.9% isotonic saline. Based on the evidence, the injectates might have less impact on the treatment efficacy.

The therapeutic effects of TPIs can be attributed to the pharmacological effects of the injectate or the activation of the reflex mechanism. Certainly, isotonic saline may be associated with the latter mechanism. Here, the gate-control theory can also be another explanation, although it is controversially discussed. According to the hypothesis presented by Simmons, calcium ions released from the damaged sarcoplasmic reticulum provoke biochemical reactions, leading to uncontrolled muscular contraction and excessive metabolites sensitizing the nociceptors. Injecting any liquid, such as isotonic saline, can wash away calcium and nerve-irritating substances, consequently, diminishing the excessive muscular contractions and nerve hyperirritability [26]. Although the analgesic mechanisms attributable to saline have not yet been elucidated, several clinical studies have suggested that the effectiveness of TPI is based on the mechanical effect of the needle and the injectate rather than its pharmacological effect [25,27,28,29,30,31,32]. Considering our findings, TPIs using isotonic saline in patients with CTTH may have a therapeutic effect. Notably, as the number of injections increased, the headache symptoms tended to further improve in our study. Therefore, saline injections at the trigger points may be used as a long-term treatment alternative for CTTH. Future studies are indisputably required to investigate the optimum number and interval of injections.

This study has several limitations. First, it is a retrospective study and lacks a control group; therefore, comprehensive data collection may be challenging. Second, the number of patients was limited, and further validation via large-scale studies is warranted. Third, as our cohort comprised only patients with CTTH, the study results cannot be generalized to all patients with different headache syndromes. Therefore, studies should be performed on patients with episodic TTH, migraine, and cluster headache.

## 5. Conclusions

This study demonstrates that the administration of new targeted TPIs using isotonic saline into the head and neck muscles could be effective in relieving headache intensity and improving the quality of life in patients with CTTH. Our results need to be further validated by conducting studies with larger sample sizes, in which this safe and cheap intervention can be comparatively assessed for different headache types, injectates, and injection protocols.

## Figures and Tables

**Figure 1 jcm-11-05428-f001:**
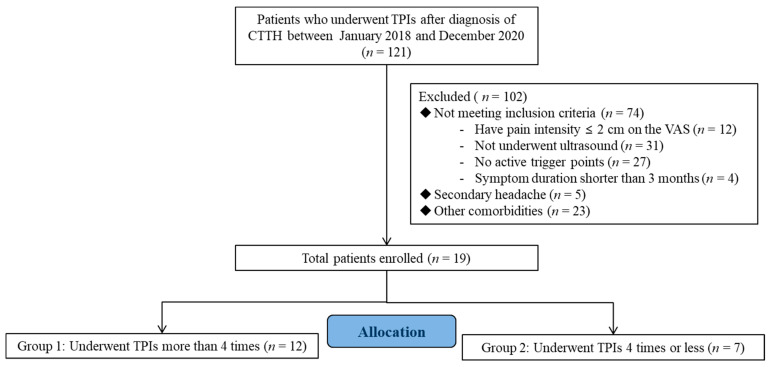
Flow diagram showing the study protocol. TPIs, trigger-point injections, CTTH, chronic tension-type headache.

**Figure 2 jcm-11-05428-f002:**
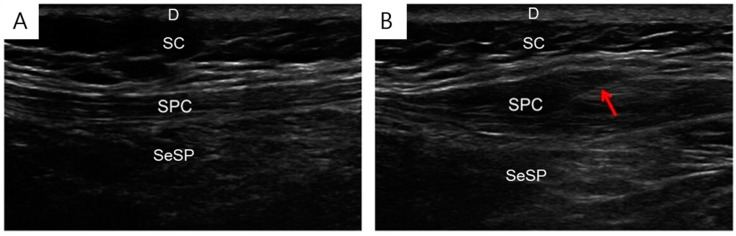
Isotonic saline injection under ultrasound guidance. Ultrasound images before (**A**) and during (**B**) the isotonic saline injection into the affected muscle. D, dermis; SC, subcutaneous; SPC, splenius capitis; SeSP, semispinalis capitis, needle (arrow).

**Figure 3 jcm-11-05428-f003:**
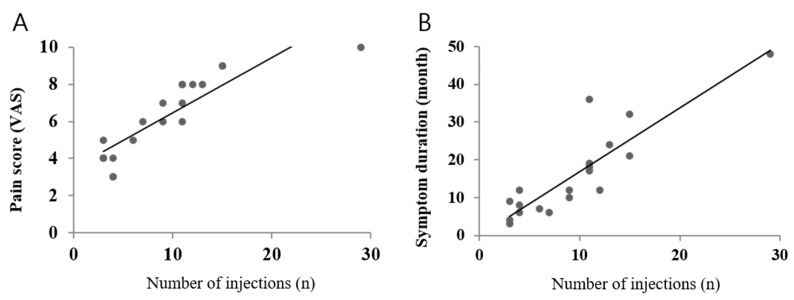
Relationship between pain score and the number of injections (**A**) and between symptom duration and the number of injections (**B**) analyzed via simple linear regression.

**Table 1 jcm-11-05428-t001:** Comparison of demographic and clinical characteristics of the patients.

Characteristic	Group 1 (*n* = 12)	Group 2 (*n* = 7)	*p* Value
Age, years, mean (range)	64.9 (23–74)	46.8 (38–81)	0.736
Sex, male/female	6/6	3/4	0.802
Symptom duration, months, mean ± SD	18.7 ± 12.6	12.0 ± 10.3	0.791
Number of injections into the muscles			
Splenius capitis	177	38	0.554
Upper trapezius	90	20	0.698
Middle trapezius	71	18	0.565
Paravertebral muscles	42	13	0.798
Scalenus medius	15	5	0.609
Levator scapulae	15	4	0.579
Sternocleidomastoid	58	10	0.627
Occipitalis	2	1	0.773
Temporalis	16	2	0.691

<0.05 calculated using the Mann–Whitney U test or chi-squared test. SD, standard deviation.

**Table 2 jcm-11-05428-t002:** Comparison of visual analog scale and headache disability inventory scores.

Variables	Group 1 (*n* = 12)	Group 2 (*n* = 7)	*p* Value
VAS (cm), mean ± SD			
Baseline	7.33 ± 2.14	6.43 ± 2.07	0.705
1 month	5.25 ± 1.96 ^‡^	2.86 ± 1.57 ^‡^	0.001 ^†^
Last injection	2.83 ± 1.80 *^,§^	2.86 ± 1.57 *^,§^	0.634
Δ (1 month–baseline)	2.08 ± 1.30	3.57 ± 1.92	0.029 ^†^
Δ (Last injection–baseline)	4.40 ± 1.54	3.57 ± 1.92	0.498
HDI (score), mean ± SD			
Baseline	63.67 ± 28.38	49.86 ± 23.61	0.956
1 month	50.85 ± 24.65 ^‡^	22.71 ± 18.64 ^‡^	0.001 ^†^
Last injection	29.83 ± 19.40 *^,§^	22.71 ± 18.64 *^,§^	0.553
Δ (1 month–baseline)	12.82 ± 14.55	27.15 ± 18.94	0.001 ^†^
Δ (Last injection–baseline)	33.84 ± 18.29	27.15 ± 18.94	0.429

VAS, visual analog scale; HDI, headache disability inventory; Δ, differences in VAS or HDI scores at baseline, 1 month after the first injection, and immediately after the final injection. Group 1, patients who received TPIs more than four times; Group 2, patients who received TPIs less than, or equal to, four times. * *p* < 0.05 calculated using Wilcoxon signed-rank test between baseline and after the final injection. ^‡^
*p* < 0.05 calculated using Wilcoxon signed-rank test between baseline and at 1 month after the first injection. ^§^
*p* < 0.05 calculated using Wilcoxon signed-rank test between 1 month after the first injection and after the final injection. ^†^
*p* < 0.05 calculated using Mann–Whitney U test between the two groups.

## Data Availability

All data generated/analyzed and used to support the findings of this study are included within the article.

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
