# Peer review of "A Pilot Analysis on the Efficacy of Multiple Trigger-Point Saline Injections in Chronic Tension-Type Headache: A Retrospective Observational Study"

_jcm, 2022, doi:10.3390/jcm11185428_

Round 1
Reviewer 1 Report
How to explain the mechanism of saline treatment for chronic tension type headache?
Author Response
8 September 2022
Dear Editor:
Thank you for your interest in our manuscript and for providing enriching suggestions.
Indeed, the comments were greatly helpful to improve the content of our manuscript. We have listed the changes below and also highlighted them in the manuscript. We hope that these revisions meet with your approval.
Sincerely yours,
Dong Rak Kwon, MD, PhD
Department of Rehabilitation Medicine
Catholic University of Daegu School of Medicine
33 Duryugongwon-ro 17-gil, Nam-Gu, Daegu, Korea, 705-718
Phone: +82 53 650 4687, Fax: +82 53 622 4687
E-mail: coolkown@cu.ac.kr
Response to Reviewer 1
Comment 1: How to explain the mechanism of saline treatment for chronic tension type headache?
Response 1: Thank you for your question. As we mentioned in the discussion, injecting isotonic saline can wash away the calcium and nerve-irritating substances, leading to reduction in excessive muscular contraction and nerve hyperirritability. We have revised the text as follows with the addition of new references:
Although the analgesic mechanisms attributable to saline have not yet been elucidated, several clinical studies have suggested that the effectiveness of TPI is based on the mechanical effect of the needle/injectate rather than its pharmacological effect [24, 26-31].
- Tschopp KP, Gysin C, “Local injection therapy in 107 patients with myofascial pain syndrome of the head and neck,” ORL J Otorhinolaryngol Relat Spec, vol. 58, no. 6, pp. 306-310, 1996.
- Lewit K, “The needle effect in the relief of myofascial pain,” Pain, vol. 6, no. 1, pp.83-90, 1979.
- Ay S, Evcik D, Tur BS, “Comparison of injection methods in myofascial pain syndrome: a randomized controlled trial,” Clin Rheumatol, vol. 29, no. 1, pp. 19-23, 2010.
- Ojala T, Arokoski JP, Partanen J, “The effect of small doses of botulinum toxin a on neck-shoulder myofascial pain syndrome: a double-blind, randomized, and controlled crossover trial,” Clin J Pain, vol. 22, no. 1, pp. 90-96, 2006.
- Frost FA, Jessen B, Siggaard-Andersen J, “A control, double-blind comparison of
mepivacaine injection versus saline injection for myofascial pain,” Lancet, vol. 1, no. 8167, pp. 499-500, 1980.
- Garvey TA, Marks MR, Wiesel SW, “A prospective, randomized, double-blind evaluation of trigger-point injection therapy for low-back pain,” Spine, vol. 14, no. 9, pp. 962-964, 1989.
- Cummings TM, White AR, “Needling therapies in the management of myofascial trigger point pain: a systematic review,” Arch Phys Med Rehabil, vol. 82, no. 7, pp. 986-992, 2001.
- Discussion
Injecting any liquid such as isotonic saline can wash away calcium and nerve-irritating substances, consequently diminishing the excessive muscular contractions and nerve hyperirritability [25]. Although the analgesic mechanisms attributable to saline have not yet been elucidated, several clinical studies have suggested that the effectiveness of TPI is based on the mechanical effect of the needle/injectate rather than its pharmacological effect [24, 26-31]. Considering our findings, TPIs with isotonic saline in patients with CTTH may have a therapeutic effect.

Reviewer 2 Report
Dear Author,
The purpose of the present study was to investigate the efficacy of new target trigger point injections and has reported that TPI is effective in reducing the pain. I really appreciate the he authors for evaluating the effect of this intervention in chronic tension HA as this is one of the most common cases of hospital visit. However, few suggestions to improve the quality of the article are given below:
1. As the sample size of the study is less, the title may be framed as "A pilot analysis on the Efficacy of Multiple Trigger Point Saline Injections in Chronic Tension-Type Headache: A Retrospective Observational Study"
2. The author have compared two groups who received TPI based on frequency. The study could have identified matched controls and compare the TPI group with control group.
3. Kindly add about the comparison of frequency of TPIs in introduction.
4. Smaller sample size may question the statistical validity of the result.
5. The flow diagram appears to be that of a prospective study. Kindly modify it.
6. The authors have used parametric test to compare both the groups. Kindly do the analysis with non-parametric tests which would improv the statistical validity.
- Kindly add this reference: Robbins MS, Kuruvilla D, Blumenfeld A, Charleston L 4th, Sorrell M, Robertson CE, Grosberg BM, Bender SD, Napchan U, Ashkenazi A; Peripheral Nerve Blocks and Other Interventional Procedures Special Interest Section of the American Headache Society. Trigger point injections for headache disorders: expert consensus methodology and narrative review. Headache. 2014 Oct;54(9):1441-59. doi: 10.1111/head.12442. Epub 2014 Aug 28. PMID: 25168295.
Author Response
8 September, 2022
Dear Editor:
Thank you for your interest in our manuscript and for providing enriching suggestions.
Indeed, the comments were greatly helpful to improve the content of our manuscript. We have listed the changes below and also highlighted them in the manuscript. We hope that these revisions meet with your approval.
Sincerely yours,
Dong Rak Kwon, MD, PhD
Department of Rehabilitation Medicine
Catholic University of Daegu School of Medicine
33 Duryugongwon-ro 17-gil, Nam-Gu, Daegu, Korea, 705-718
Phone: +82 53 650 4687, Fax: +82 53 622 4687
E-mail: coolkown@cu.ac.kr
Response to Reviewer 2
Comment 1: As the sample size of the study is less, the title may be framed as "A pilot analysis on the Efficacy of Multiple Trigger Point Saline Injections in Chronic Tension-Type Headache: A Retrospective Observational Study"
Response 1: Thank you for your suggestion. We have revised the title of our study accordingly.
Comment 2: The author have compared two groups who received TPI based on frequency. The study could have identified matched controls and compare the TPI group with control group.
Response 2: As this was a retrospective study, recruiting an appropriate control group was not possible.
Comment 3: Kindly add about the comparison of frequency of TPIs in introduction.
Response 3: Based on you comment, we have revised the Introduction as follows:
- Introduction
Therefore, the purpose of this study was to investigate the efficacy of new target TPIs using isotonic saline, which does not elicit side effects with repeat injections. We also aimed to clarify the relationship among clinical features, CTTH symptoms, and the frequency of TPIs.
Comment 4: Smaller sample size may question the statistical validity of the result.
Response 4: To increase the statistical power and minimize sample heterogeneity, we applied strict inclusion and exclusion criteria. Additionally, when planning the study, we focused on the reliability of measurements using various clinical scales (e.g., visual analog scale and Henry Ford Hospital Headache Disability Inventory). As we mentioned in the Discussion, a larger sample size and a longer follow-up period will be needed to validate the clinical efficacy reported in this study.
Comment 5: The flow diagram appears to be that of a prospective study. Kindly modify it.
Response 5: We have revised the flow diagram accordingly.
Comment 6: The authors have used parametric test to compare both the groups. Kindly do the analysis with non-parametric tests which would improve the statistical validity.
Response 6: Thank you for your comment. We reanalyzed our results using nonparametric tests. We have revised the pertinent parts in the Methods, Results, and Tables as follows:
2.5. Statistical analysis
IBM SPSS ver. 19.0 (IBM Co., Armonk, NY, USA) was used for statistical analyses, and significance was indicated by P < 0.05. The chi-squared test, Mann–Whitney U-test independent t test, and Wilcoxon signed-rank test paried t test were used to determine the inter- and intragroup statistical differences.
Table 1. Comparison of demographic and clinical characteristics of the patients
|
Characteristics |
Group 1 (n = 12) |
Group 2 (n = 7) |
P value |
|
Age, years, mean [range] |
64.9 [23–74] |
46.8 [38–81] |
0.736 |
|
Sex, male/female |
6/6 |
3/4 |
0.802 |
|
Symptom duration, months, mean ± SD |
18.7 ± 12.6 |
12.0 ± 10.3 |
0.791 |
|
Number of injections into the muscles |
|
|
|
|
Splenius capitis |
177 |
38 |
0.554 |
|
Upper trapezius |
90 |
20 |
0.698 |
|
Middle trapezius |
71 |
18 |
0.565 |
|
Paravertebral muscles |
42 |
13 |
0.798 |
|
Scalenus medius |
15 |
5 |
0.609 |
|
Levator scapulae |
15 |
4 |
0.579 |
|
Sternocleidomastoid |
58 |
10 |
0.627 |
|
Occipitalis |
2 |
1 |
0.773 |
|
Temporalis |
16 |
2 |
0.691 |
*P < 0.05 calculated using the Mann–Whitney U test or chi-squared test. SD, standard deviation
Table 2. Comparison of visual analog scale and headache disability inventory scores
|
Variables |
Group 1 (n = 12) |
Group 2 (n = 7) |
P value |
|
VAS (cm), mean ± SD |
|||
|
Baseline |
7.33 ± 2.14 |
6.43 ± 2.07 |
0.705 |
|
1 month |
5.25 ± 1.96‡ |
2.86 ± 1.57‡ |
0.001† |
|
Last injection |
2.83 ± 1.80*§ |
2.86 ± 1.57*§ |
0.634 |
|
Δ(1 month–baseline) |
2.08 ± 1.30 |
3.57 ± 1.92 |
0.029† |
|
Δ(Last injection–baseline) |
4.40 ± 1.54 |
3.57 ± 1.92 |
0.498 |
|
HDI (score), mean ± SD |
|||
|
Baseline |
63.67 ± 28.38 |
49.86 ± 23.61 |
0.956 |
|
1 month |
50.85 ± 24.65‡ |
22.71 ± 18.64‡ |
0.001† |
|
Last injection |
29.83 ± 19.40*§ |
22.71 ± 18.64*§ |
0.553 |
|
Δ(1 month–baseline) |
12.82 ± 14.55 |
27.15 ± 18.94 |
0.001† |
|
Δ(Last injection–baseline) |
33.84 ± 18.29 |
27.15 ± 18.94 |
0.429 |
VAS: visual analogue scale; HDI: headache disability inventory; Δ: differences in VAS or HDI scores at baseline, 1 month after the first injection, and immediately after last injection. Group 1, patients who received TPIs more than four times; Group 2, patients who received TPIs less than or equal to four times.
*P < 0.05 calculated using Wilcoxon signed-rank test independent t test between baseline and after the last injection.
‡P < 0.05 calculated using Wilcoxon signed-rank test independent t test between baseline and at 1 month after the first injection.
- P < 0.05 calculated using Wilcoxon signed-rank test independent t test between 1 month after the first injection and after the last injection.
†P < 0.05 calculated using Mann–Whitney U paried t testtest between the two groups.
Comment 7: Kindly add this reference: Robbins MS, Kuruvilla D, Blumenfeld A, Charleston L 4th, Sorrell M, Robertson CE, Grosberg BM, Bender SD, Napchan U, Ashkenazi A; Peripheral Nerve Blocks and Other Interventional Procedures Special Interest Section of the American Headache Society. Trigger point injections for headache disorders: expert consensus methodology and narrative review. Headache. 2014 Oct;54(9):1441-59.
Response 7: We have cited the reference (14) in the Introduction and added it to the References.
